# Planning a Collection of Virtual Patients to Train Clinical Reasoning: A Blueprint Representative of the European Population

**DOI:** 10.3390/ijerph19106175

**Published:** 2022-05-19

**Authors:** Anja Mayer, Vital Da Silva Domingues, Inga Hege, Andrzej A. Kononowicz, Marcos Larrosa, Begoña Martínez-Jarreta, Daloha Rodriguez-Molina, Bernardo Sousa-Pinto, Małgorzata Sudacka, Luc Morin

**Affiliations:** 1Medical Education Sciences, University of Augsburg, 86159 Augsburg, Germany; inga.hege@med.uni-augsburg.de; 2School of Medicine and Biomedical Sciences, University of Porto, 4050-513 Porto, Portugal; vitalmsd@gmail.com; 3Department of Bioinformatics and Telemedicine, Jagiellonian University Medical College, 30-688 Krakow, Poland; andrzej.kononowicz@uj.edu.pl; 4Aragón Health Research Institute (IIS-Aragón), University of Zaragoza, 50009 Zaragoza, Spain; marcoslarrosa59@gmail.com (M.L.); mjarreta@unizar.es (B.M.-J.); 5Institute and Clinic for Occupational, Social and Environmental Medicine, University Hospital, LMU Munich, 80336 Munich, Germany; daloha.rodriguez_molina@med.uni-muenchen.de; 6MEDCIDS—Department of Community Medicine, Information and Health Decision Sciences, Faculty of Medicine, University of Porto, 4200-319 Porto, Portugal; bernardo@med.up.pt; 7Department of Medical Education, Jagiellonian University Medical College, 30-688 Krakow, Poland; malgorzata.sudacka@uj.edu.pl; 8Pediatric and Neonatal Intensive Care Unit, DMU 3 Santé de L’enfant et de L’adolescent, APHP Paris Saclay, Bicêtre Hospital, 94270 Le Kremlin-Bicêtre, France; luc.morin@aphp.fr; 9Institute of Integrative Biology of the Cell, CNRS, CEA, Paris Saclay University, 91190 Gif-sur-Yvette, France

**Keywords:** clinical reasoning, medical education, international collaboration, virtual patients, case-based learning, open educational resources

## Abstract

Background: Virtual patients (VPs) are a suitable method for students to train their clinical reasoning abilities. We describe a process of developing a blueprint for a diverse and realistic VP collection (prior to VP creation) that facilitates deliberate practice of clinical reasoning and meets educational requirements of medical schools. Methods: An international and interdisciplinary partnership of five European countries developed a blueprint for a collection of 200 VPs in four steps: (1) Defining the criteria (e.g., key symptoms, age, sex) and categorizing them into disease-, patient-, encounter- and learner-related, (2) Identifying data sources for assessing the representativeness of the collection, (3) Populating the blueprint, and (4) Refining and reaching consensus. Results: The blueprint is publicly available and covers 29 key symptoms and 176 final diagnoses including the most prevalent medical conditions in Europe. Moreover, our analyses showed that the blueprint appears to be representative of the European population. Conclusions: The development of the blueprint required a stepwise approach, which can be replicated for the creation of other VP or case collections. We consider the blueprint an appropriate starting point for the actual creation of the VPs, but constant updating and refining is needed.

## 1. Introduction

Virtual patients (VPs) are “interactive computer simulation[s] of real-life clinical scenarios for the purpose of medical training, education, or assessment” [1]. Their importance has been increasing over the years [2], especially since the beginning of the COVID-19 pandemic [3]. VPs provide a safe environment in which learners can work at their own pace and learn from errors without harming a patient [4]. They are typically designed to unfold in a step-by-step manner, revealing the information about a patient over time leading the learner to the final diagnosis [5,6]. During the process, multimedia elements add authenticity to the VP scenario. Overall, several studies indicated that VPs have the potential to train students in clinical reasoning [7,8,9].

Clinical reasoning is a complex process in which healthcare professionals (e.g., physicians, physiotherapists, nurses) gather and interpret information, generate hypotheses, derive a final diagnosis, and develop treatment plans [10]. VPs demonstrated their effectiveness in improving components of clinical reasoning such as data gathering, generating differential and final diagnoses, and developing a treatment plan [11,12]. However, this process is not only influenced by medical knowledge, but, according to situativity theory, also by contextual factors related to the patient and to the encounter [13,14]. Such factors can be the setting (e.g., emergency room, general practice), or the patient’s age, sexual orientation, comorbidities [15], or behavior [5].

Therefore, providing a collection of VPs to medical students offers the potential for deliberate training of clinical reasoning, but the careful selection of these contextual factors, key symptoms, and diagnoses is crucial [16,17]. Such a balanced selection of key symptoms and (differential) diagnoses in a VP collection prepares students for situations they are likely to experience in practice [18]. Moreover, it enables them to train clinical reasoning by comparing and contrasting, i.e., when they face cases with similar clinical findings, they have to weigh different options based on the relative probability of each diagnosis and the typicality of findings [10,19]. Deliberately varying contextual factors and including atypical presentations influences the complexity of a VP [10,19]. Furthermore, contextual factors are important to create realistic and authentic scenarios [20] that represent the diversity of a patient population adequately. An under- or non-representation of marginalized groups holds the danger of an unintended hidden curriculum [21]. i.e., unintended messages that can bias students, who for example might be subconsciously trained to perceive patients that are male, heterosexual, white, and cis (i.e., the opposite of transgender) as the standard patients in Western countries [22,23,24].

However, previous studies showed that existing VP or case collections tend to represent the real world only to a limited extent in terms of key symptoms, diagnoses, and contextual factors [24,25,26]. For example, one case collection [25] scarcely included patients with a disability, migration background, or chronic conditions despite their worldwide relevance in healthcare [27]. Although resources and didactical advice for the creation of individual VPs are available [18,28], there is hardly any guidance for designing a collection of VPs. Previous VP collections [29] and projects such as [30,31] did either not follow or publish a sophisticated planning approach.

To address this shortcoming on an international level, we formed an interdisciplinary partnership of six institutions from Poland, Germany, Spain, Portugal, and France. In our project, iCoViP (international Collection of Virtual Patients) [32], we aimed at planning and delivering a diverse and realistic VP collection that facilitates deliberate practice of clinical reasoning and meets educational requirements of medical schools in Europe. Therefore, our objectives were to develop and assess a reusable process that helps medical educators plan or evaluate VP collections ensuring that they are

(1)realistic in terms of patient characteristics in Europe,(2)aligned to educational objectives of medical curricula in Europe to ease curricular integration, and(3)suitable to train clinical reasoning by comparing and contrasting similar cases in varying contexts.

## 2. Materials and Methods

Due to the lack of guidance on how to implement such a process, we based our planning on our experience from previous projects [25,30,31] and developed a four-step approach to create the blueprint for our VP collection (see Figure 1). Consequently, this blueprint will serve as the basis for the subsequent VP creation. As agreed upon in the grant proposal of the project, we aimed at creating 125 new VPs to extend an existing collection of 75.

Step 1: Definition of criteria for the blueprint

As a first step, we defined the criteria, i.e., variables which needed to be specified for all VPs. We started by extracting criteria from the literature and the experience of our consortium to describe the VPs in the blueprint. Afterwards, we discussed these criteria with all partners and grouped them into (i) disease-related, (ii) patient-related, (iii) learner-related, and (iv) encounter-related criteria, which can be briefly described as followed:

(i) Disease-related: These criteria include the final diagnoses and key symptoms, i.e., the complaints that are the primary reasons for the VPs’ visit. This ensures that our collection will cover the most common diseases and symptoms as the main aspects of the clinical reasoning process. We mapped the final diagnoses to the Medical Subject Heading (MeSH) [33] and the national competency frameworks of partner countries which describe the learning outcomes to be covered by medical schools [34,35,36,37,38]. The mapping with the intended learning outcomes and a standard biomedical thesaurus like MeSH facilitates a curricular integration of VPs covering specific learning objectives. Additionally, we clustered the final diagnoses in disease groups based on their pathogenetic pathway, similarly to the VINDICATE approach used in medical education [39]. We also agreed on covering the onset of the disease, e.g., acute, or chronic, to adequately represent the importance of chronic diseases [27] and avoid their underrepresentation. Finally, we included the closure of the scenario, i.e., whether a patient dies or is successfully discharged, to provide a realistic outcome and avoid tabooing the dying of patients.

(ii) Patient-related: These criteria are the VPs’ characteristics, including their age, sex, sexual orientation, profession, ethnicity, cultural or migration background, disability, and addiction/substance abuse. These aspects are crucial to ensure a diverse and authentic patient population [24,25] and to raise awareness for common biases in the clinical reasoning process [40,41,42].

(iii) Learner-related: We included the role of the learner in the VP scenario as a criterion, as this is an important factor in simulation-based environments [43]. The learner is cast for instance in the role of a resident or consultant, being responsible for a patient.

(iv) Encounter-related: We included the setting in which the consultation with the VPs takes place, such as a university hospital or a doctor’s office, as a criterion. In doing so, we aimed at avoiding an overrepresentation of well-equipped university hospitals and emergency departments and at providing a realistic variety of facilities. Table 1 provides an overview of all included criteria. Once we agreed on all the criteria and the permitted values for list selection under each criterion, we developed a template that allowed partners to populate the blueprint with VP data.

Step 2: Identification of data sources for assessing the representativeness of the blueprint.

All partners agreed on a list of suitable key symptoms that the VPs should present with; these were extracted from the literature [44,45] and available national competency frameworks [34,35,36,37,38].

To identify frequent diagnoses, we based our literature search on a systematic review by Finley et al. that reports on the most common conditions in primary care [46]. Additionally, we included articles that recommend diagnoses to be covered in medical education from a daily-practice perspective [47,48,49].

To assess whether our populated blueprint is, as intended, a realistic representation of the European patient population, we identified the most recent, pertinent data from the World Health Organization (WHO) and statistics available for Europe to compare to our VPs.

All selected references and data sources are included in Table 1.

Step 3: Populating the blueprint

In this step, we mapped the existing 75 VPs into our blueprint and asked each partner institution to add 25 VPs they intended to develop based on their curricular needs.

Step 4: Refinement and reaching consensus

First, we compared the results of step 3 (i.e., the distributions of suggested patient-, learner-, and encounter-related criteria of the blueprint) to the identified literature and statistics (see step 2). Then, if necessary and feasible, we asked partner institutions to modify their VP scenarios to represent a more realistic patient population.

To foster students’ comparing and contrasting of similar cases, which is vital to build competency in clinical reasoning, we excluded key symptoms that were only present in one VP from the final blueprint. Therefore, adjustments had to be made in some cases, leading to a reduced number of key symptoms in the final blueprint.

As the diagnoses of the initially outlined VPs differed substantially from the results of the literature, we applied a Delphi-like approach to reach consensus on the final diagnoses (for details see Appendix A). This approach is widespread in medical education research [62] and curriculum planning [63,64].

Once the final blueprint was approved by all partners, we mapped all VPs to MeSH and the national competency frameworks and performed descriptive analysis for all criteria.

## 3. Results

Our three main results are the populated blueprint describing the VPs, which has been published as part of the project [65], the descriptive statistics of the criteria, and the actual decision-making process. In the following sections, we show the results of our descriptive analyses.

### 3.1. Analysis of Disease-Related Criteria

Our final list of VPs includes 29 common key symptoms [65]. The Delphi-like approach resulted in a total of 176 diagnoses across several healthcare disciplines displayed in our blueprint. Of these, 22 are covered by more than one VP [65].

The final diagnoses of our VPs include all 11 main disease categories of the German national catalog of learning objectives in medicine (NKLM) [34], and 15 of 17 disease categories listed in the national framework published by the Polish Ministry of Science [35]. The blueprint also covers 17 of 18 categories in the whitepaper published by the Spanish National Agency for Quality and Accreditation [37], and 11 of 12 categories listed in the French catalog of learning objectives [38]. The Portuguese national framework consists of 5 main groups of diseases that are all covered by our VPs [36]. Although these frameworks vary in their structure and content, they mostly focus on diseases of the cardiovascular, digestive, and respiratory systems. The missing categories were particular topics, such as “safe use of medication” in the French learning objective catalog, or specific disciplines, such as ophthalmology in the Spanish and the Polish frameworks.

Regarding the disease groups of the VINDICATE mnemonic, the most frequent groups were “infectious” (31%), “vascular” (14%) and “immunologic” (14%) (for further values see Appendix B, Table A1).

The VP’s onset of symptoms was acute, subacute, or chronic for 62%, 19%, and 19%, respectively. At the end of the scenario, 6% of the VPs die, 36% require long-term treatment, and 59% are successfully discharged.

### 3.2. Analysis of Patient-Related Criteria

Our blueprint of 200 VPs includes all age groups, from newborns up to 93 years, with an age distribution of 12% between 0–14, 61% between 15 and 64, and 28% above 65. Compared to the general population in Europe in 2021 (15% between 0–14, 64% between 15 and 64, and 21% above 65) [50], our VPs are older, considering that elderly people tend to seek medical care more frequently than the average population.

The final blueprint includes 1% intersexual or transgender patients compared to an estimated rate of up to 2% in European adults [52]. Eight percent of our VPs are described to be homo-or bisexual, which is slightly higher than the population in Europe, in which the lesbian, gay, bisexual, and transgender (LGBT) population varies among countries but is reported highest in Germany with over 7% in 2016 [53].

Regarding ethnicity, our blueprint comprises White (89%), Black (8%), Asian (2%), and Hispanic (2%) VPs. However, we did not find suitable European data sources for comparison.

The professions of the VPs include 27 of the 38 occupational sectors covered in the Eurostat employment statistics [55] with the five most common occupations being teaching professionals, health professionals, researchers & engineers, personal service workers, and sales workers. The most common occupations in the EU are sales workers, office associate professionals, personal service workers, teaching professionals, and drivers & vehicle operators [55]. The proportion of unemployed VPs among the working-age population is 6%, a bit lower than the average seasonally adjusted unemployment rate in the EU, which ranged between 7 and 8% in 2020 and 2021 [56].

In our blueprint, we have 8% of disabled VPs, compared to an estimated prevalence of 6 to 10% for disabilities in Europe [57].

We included 13% of VPs with a relevant cultural or migration background, for example, refugees or VPs with language barriers. In the EU, the proportion of non-EU citizens and people that were born outside of the EU was 5% and 8%, respectively, in 2020 [58].

In the age group over 15 years, 17% of the VPs are smokers, compared to an average of 18% daily smokers in 2019 in the EU [59]. Moreover, 8% of the VPs in this age group have a relevant alcohol consumption which corresponds to the proportion of adults in the EU with daily alcohol consumption [60].

Table 2 provides a summary of patient-related criteria; all criteria are presented in detail in Table A1 (Appendix B).

### 3.3. Analysis of Encounter- and Learner-Related Criteria

The learner role is in 46% of the scenarios a resident, in 30% a consultant, and in 18% an intern (for more details see Appendix B, Table A1). About one-third (34%) of the encounters occur in a medical practice, 23% in a hospital, 21% in a university hospital, 12% in an outpatient clinic, and 11% in a rural hospital. Of all hospital settings, 31% take place in an emergency department. 

## 4. Discussion

We developed a blueprint for a VP collection that covers a wide range of medical education needs while ensuring a realistic degree of diversity. The approach we followed to develop this blueprint was challenging and time-consuming. Nevertheless, we believe that it was worth the effort as it enabled us to meet our stated objectives.

To reach our first objective of being realistic in terms of patient characteristics at an international level, we compared the VPs described in our blueprint to sources of population statistics, such as WHO or EUROSTAT. We did not rely on randomizing clinical data for the development of the blueprint [66] but rather reached consensus through a cycle of group discussions, a modified Delphi-approach, and verification steps. Overall, our analyses show that the VPs described in the blueprint represent a patient population more diversely and realistically when compared to previous initiatives [24,25,67]. However, despite our best efforts, we still have some deviations we will refine further during the VP development. For example, the VPs cover a wide range of occupations, including unemployed and retired people, but we have an overrepresentation of professions our consortium is familiar with, such as health and teaching professionals. Also, the proportion of disabled VPs and chronic onsets is lower than we intended it to be. On the other hand, about one-third of our VPs require long-term treatment, emphasizing the importance of chronic conditions in healthcare. We will keep these aspects in mind during the actual development of VPs and refine the collection accordingly.

Our second objective was to ensure the applicability and integrability of the VP collection for medical schools in Europe. To reach this objective, we considered different perspectives from health professions, educators, and researchers across Europe, similar to previous international and interdisciplinary approaches [30,31]. Thus, we were able to consider cultural differences of involved countries [68] as well as country-specific diagnostic and therapeutic approaches [69]. Additionally, we mapped the VPs in our blueprint to national competency frameworks. Through this approach, we hope to ease the integration of the VP collection into the curricula of other schools and increase its acceptance by healthcare educators in Europe [70,71,72].

Our third objective was to plan a VP collection that is suitable to deliberately practice clinical reasoning by comparing and contrasting similar case presentations. There is no gold-standard available for the optimal composition of a VP collection in terms of key-symptoms, diagnoses, and contextual variation. However, Kassirer et al. pointed out that learning clinical reasoning is fostered by a larger collection of VPs [73]. As we based our approach on available guidance as far as possible [28,74], we believe that covering a range of 29 key-symptoms and 176 final diagnoses is a good starting point, but we also see the need for further expanding the collection in the future. Looking at teaching of clinical reasoning through the lenses of situativity theory [13] emphasizes the importance of the context in making diagnostic and therapeutic decisions. For example, the diversity of the VP characteristics exposed in our blueprint provides a valuable opportunity to provide thoughtful feedback to students on how such factors influence clinical reasoning and can be the source of cognitive errors. A discussion or review of errors is a suitable approach to improve clinical reasoning as long as it includes elaborated feedback [75,76].

In addition to reaching our objectives, our discussions helped us in developing a better understanding of patient- and healthcare-related aspects in other countries, which will enrich our teaching and research. For instance, we had a vivid discussion on defining migration background and differences in countries of origin between our partner countries.

Finally, we believe that our approach emphasizes the importance of creating VP or case collections that are diverse, realistic, as well as applicable and shareable on an international level. We hope that implementing such a VP collection in study programs will contribute to their internationalization, which has the potential to facilitate the mobility of health professionals in Europe.

We are aware that our approach has several limitations.

First, we could not identify data sources based on the patient population in Europe to compare our patient-related criteria. Therefore, we had to use statistical data based on the general population in Europe, which certainly differs in terms of age distribution. In addition, we had to obtain information from various data sources, which may have used different methodologies and underlying definitions. However, our aim was not to simulate a perfect real-world patient population but to consider aspects of diversity and an approximation to a real-world population.

Second, we did not yet perform sub-analyses on the distributions involving the combination of different criteria themselves (for example, how disability status or migration background are distributed in relation to the VPs’ sex). During the VP development, further analyses will be undertaken to overcome gender- and ethnicity-related stereotypes.

Furthermore, despite having a diverse group of partners in the project, we must acknowledge that for instance no Scandinavian or Balkan country is represented in our group and therefore, we might be missing the perspective of Northern or Southeastern European countries.

The experience of the recent COVID-19 pandemic taught us that the global health situation can change dramatically in a very short time frame. Global warming, political conflicts, economic crises, but also accelerating technical innovation are likely to contribute to increased fluctuations in patient populations characteristics. Our paper adds to the existing body of knowledge by proposing a methodology to help educators to efficiently reach a realistic VP or case mix suitable for clinical reasoning training. We foresee that there will be a growing demand for such methods to make the educational system fit the actual needs of the changing society. At the same time, we stress that the authenticity reached by such a methodology is not given for ever and that revisions and updates of VP collections are required to keep them up to date.

## 5. Conclusions

In this study, we developed a process for designing a blueprint for a VP collection that provides a realistic picture of the European patient population, meets educational requirements, and facilitates deliberate practice of clinical reasoning. We believe that our work can help healthcare educators to assess their own VP or case collections, expand their collections within their schools (or in collaboration with others), or implement a similar process. Despite some limitations, we think that the blueprint, which is the product of this process, is an appropriate starting point for creating the VPs. The dynamic nature of the pandemic situation will probably make it inevitable that we will have to do further adaptations and refinements during the creation phase. Also, we consider the blueprint a dynamic document we will regularly consult and update during and after creating the VPs to preserve the high quality of our VP collection.

## Figures and Tables

**Figure 1 ijerph-19-06175-f001:**
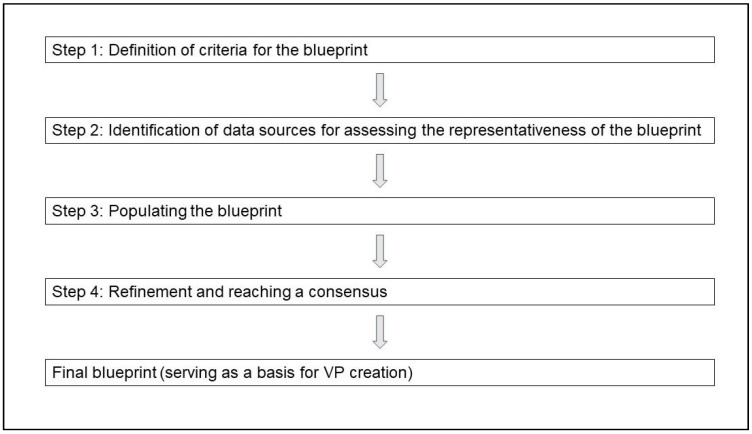
Flowchart of our four-step approach to create the blueprint. (VP = Virtual patient).

**Table 1 ijerph-19-06175-t001:** Overview of criteria, permitted values, and sources for comparison.

Criterion	Values	Sources for Comparison
Disease-related		
Key symptoms	N/A	[34,35,36,37,38,44,45]
Final diagnoses	N/A	[46,47,48,49]
Disease group	Vascular/Infectious/Neoplastic/Drugs, Toxic/Idiopathic/Congenital/Autoimmune, Immunologic/Traumatic/Endocrine, Metabolic [39]	N/A
Onset	Chronic/Subacute/Acute	N/A
Scenario closure	Long-term treatment/Successfully discharged/Died [26]	N/A
Patient-related		
Age	N/A	[50]
Sex/Gender	Female/Male/Transgender/Intersexual [24]	[51,52]
Sexual orientation	Heterosexual/Homosexual/Bisexual/Not sexually active (child)/Not stated [24]	[53]
Ethnicity	Hispanic/Black/White/Asian/Other [54]	N/A
Profession	Naming of profession/Unemployed/Retired/Student/Child/Not stated	[55,56]
Disability	Yes/No	[57]
Relevant cultural, language, or migration background	Yes/No	[58]
Addiction/substance abuse	Smoker/Ex-Smoker/Alcohol/Illegal drugs/Other/No/Not Stated [25]	[59,60,61]
Learner-related		
Learner’s role	Student/Intern/Resident/Consultant/Other/Not Stated	N/A
Encounter-related		
Setting	Rural Hospital/University Hospital/Hospital/Practice/Outpatient Clinic/Emergency Room [25]	N/A

N/A = Not applicable.

**Table 2 ijerph-19-06175-t002:** Summary of patient-related criteria and their respective data for comparison.

Criterion	Values	N	% Total	EU/European Data for Comparison	Reference
Age	0–14 years	19	11.5%	15.1%	[50]
	15–64 years	122	61.0%	64.1%	
	≥65 years	55	27.5%	20.8%	
Sex/Gender	Female	101	50.5%	51.7%	[51]
	Male	97	48.5%	48.3%	
	Transgender	1	0.5%	Estimated proportion: 0.1–2%	[52]
	Intersexual	1	0.5%		
Sexual orientation	Heterosexual	115	57.5%	N/A	
	Homosexual	12	6.0%	Proportion of LGBT: 1.5–7.4%	[53]
	Bisexual	4	2.0%		
	Not sexually active (child)	21	10.5%	N/A	
	Not stated	48	23.5%	N/A	
Disability	Yes	14	6.5%	6–10%	[57]
	No	186	93.0%	N/A	
Relevant cultural, language, or migration background	Yes	27	13.5%	5.1% non–EU citizens 8.3% born outside EU	[58]
	No	173	86.5%	N/A	
Addiction/Substance abuse *	Smoker	30	16.6% ^2^	18.4% ^1^ daily smoking	[59]
	Ex–Smoker	11	6.1% ^2^	N/A	
	Alcohol	14	7.7% ^2^	8.4% ^1^ daily alcohol	[60]
	Illegal drugs	3	2.4% ^2^	“Last month prevalence”: 1.2–8.7% ^2^	[61]
	Other	1	0.6% ^2^	N/A	
	No	111	55.0%	N/A	
	Not Stated	35	17.5%	N/A	

* Numbers do not sum up to 200 (100%) since combinations of values are possible. ^1^: Referring to age group ≥15 years. ^2^: Referring to age group 15–64 years. N/A = Not applicable.

## Data Availability

Data analyzed and generated during the present work are available in the manuscript and online: [65].

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
