# Peer review of "Planning a Collection of Virtual Patients to Train Clinical Reasoning: A Blueprint Representative of the European Population"

_ijerph, 2022, doi:10.3390/ijerph19106175_

Round 1

Reviewer 1 Report

The authors of this manuscript address a relevant and innovative topic. Virtual reality has great potential in different fields, including medicine. The exposition of the study is correct and coherent, but the following improvements are recommended: 
1. Describe in the introduction in more depth, the basic characteristics of virtual patients. This will help the reader to better understand the resource created and its possibilities.  
2. Accurately state in the introduction the previous creations of virtual patients and compare the contributions of the new virtual patients created in this study. 
3. Present objectives that will guide the study and serve to define the conclusions and findings.  
4. In the method section, cite previous studies that support the steps followed to create the virtual patients. At present, no author citations are provided to justify the steps followed. 
5. In the discussion section, include a greater number of citations that explain relationships and comparisons with other similar studies. The results presented should be compared with other results to generate links in the scientific community. 
6. Present conclusions that correspond to previously stated objectives. Likewise, in the practical implications, it is recommended to be more precise in the protocol of uses of the virtual patients created and the availability of use in medical practice and in the training stage of physicians. 

The proposed recommendations would improve the quality of the manuscript for publication and would make a greater contribution to the scientific community.

Reviewer 2 Report

Title
I found the title to be misleading. The paper actually describes the blueprinting for a collection of VPs, which have, among other things, an intention to train clinical reasoning. 

Introduction
The concept of clinical reasoning as one that can be taught is an interesting frontier in medical education without a strong evidence base. While exposure to patients is clearly an important factor in helping clinicians decide whether one diagnosis or approach has more merit than another, how exactly this happens, and to what extent all learners pick this up differently remains to be established. VPs may be useful in this process, but the gaps in the evidence base should be made clear. The reference cited (4) is a meta-analysis and discussion of the original sources that this paper refers to in its conclusions on clinical reasoning would be beneficial, especially as clinical reasoning is one of the key pillars of this submission. 
I found the approach to bring realism to the gender, ethnic origin, type of case and outcome refreshing and one i and others could learn from. 

Materials and methods
I enjoyed the systematic approach to the creation of the VP library, especially how this was explicitly linked to other sources such as diagnoses in primary care. The process was well set out and explicit. 

Results
It would seem that the process has produced a range of VPs, evidenced by the mapping onto national curricula. The explicit way that the outcomes were stated before the VPs were written was helpful for case writers. 
The authors have already published the outcome of this process (ref 54), so this paper merely describes the process of how the team arrived at them. As a full length paper, it would seem that this is a large amount of text to describe a blueprinting process and statistics around its outcome, when the important part is likely to be the product itself, which has already been published. 
Adding to this, section 3.2’s first sentence implies that the VPs have already been written. As most VP writers will know, this is the challenging part of the process. 

Discussion
VPs, which although they offer several advantages for learning, and hard to create well. The discussion brings out the value of planning explicitly what VPs should be written and how they should be integrated into the curriculum. 
It is here that clinical reasoning reappears after the introduction. I would agree a discussion of personal diagnostic biases could be augmented by tutorials centred around a library of cases as described by the authors. However, I personally  feel that either the paper needs a considerably wider discussion and justification around clinical reasoning, or it should be left out. 

Overall
The paper is well written with a good standard of English, and is very readable. The paper describes a method in great detail and its output, which is published online elsewhere. 
As a reader and VP writer, I would be far more interested in the list of cases, less in the details of the process that derived that list, so would suggest that if accepted the paper were substantially shorter than that presented in the manuscript. I would also have been frustrated that the title, while promising to assist in the delivery of clinical reasoning training, does not reflect the content of the paper. A more representative title would be “Developing an International Collection of Virtual Patients representative of European Clinical Practice”. 

Author Response

Dear reviewer,

we would like to thank you for your valid and helpful comments. Please find our corresponding answers below:

"The author describes here as health care professional gathers – can it be possible to describe more clearly who are the target health care professional 0r is it all the health care professionals. Lone 51"

Answer: Thank you to the reviewer for his excellent comments. The definition of clinical reasoning provided in the introduction refers to all health care professionals. According to this, VPs can be created and applied by different stakeholders of the healthcare system. We provided some examples in the corresponding sentence (paragraph 2). However, since our approach mainly focuses on providing a VP collection to medical students, we changed the wording accordingly in the introduction (see paragraph 3).

"Line 60 m who are these health professional students"

Answer: Please see answers for point above. 

"Line 75, I personally feel to explain or put in simple terms what do you mean by White cis men"

Answer: The term “cis” describes a person who has the same gender identity as assigned at birth, in other words, by “cis” we mean the opposite of “transgender”. We changed the sentence in the introduction (paragraph 3) accordingly for a better understanding.

"Line 106: The sentence,” These criteria include the key symptoms the VPs initially com- 106 plain about and their final diagnoses” looks little unclear, please modify the sentence"

Answer: We changed the sentence for a better understanding. The revised sentence in step 2 of the methods section is now: “These criteria include the final diagnoses and the key symptoms, i.e., the complaints that are the primary reasons for the VPs’ visit.”

Furthermore, we've got advice by a professional to improve the language and style of the manuscript.

Many thanks and kind regards!

Round 2

Reviewer 1 Report

The authors have made the suggested changes and the manuscript has been improved. 

Author Response

Dear reviewer,

We would like to thank you once again for your valuable feedback and the effort you put into the review of our manuscript.

As recommended, we had again support from a native speaker to check our manuscript concerning language and grammar and slightly adapted the manuscript accordingly.

Many thanks and best regards.

Reviewer 2 Report

Many thanks for asking me to review this again. I very much appreciated the comments by the authors in response to my own review. The changes make the paper far better. The manuscript is now a useful addition to the literature on VPs, but without overreach.

Two comments remain - it is still quite a long paper for what it says, and the journal will have to decide whether this is justified.

Second is around the justification for VPs as useful in clinical reasoning. I can't quite see how this has changed in the introduction. The key sentence "Overall, VPs are a suitable method for students to train their clinical reasoning abilities" is referenced by David Cooks excellent 2010 metanalysis of the topic. In this paper, the section on the effectiveness of VPs in clinical reasoning takes you to 5 papers, which show an outcome in favour of the VP against no instruction. I hadn't looked at these in detail, but did so to write this review. For there to be evidence of VPs promoting clinical reasoning I would have expected relevant details on the VP, the details on the assessment of reasoning, and both conforming to current understanding of these concepts.

The first paper from 1972 is set in dental education, with no abstract or paper that i could find, but suspect their idea of "Computer instruction" in 1972 cannot be the same as a VPs are now. The second from 1977 did use a computer system likely to resemble a VP, but the evaluation of decision making had just 12 students answering 6 questions, without explaining how this relates to clincal reasoning. The third, from 1986 featured 14 students exploring clinical presentations by computer, but there is no mention of clinical reasoning or decision making in the paper, just factual acquisition. The forth, in 1997 is in the field of geriatric dental hygeine. I was unable to access the full paper, but the abstract contains the phrase "clinical problem-solving skills of information gathering, assessment, and treatment" suggesting a different idea of clinical reasoning to that described in the manuscript. Lastly the fifth paper, a large multicentre study published in 2007 does review clinical reasoning. However, this is only a feature of the discussion, with no attempt made experimentally to explore the process. Rather the work focusses on inter-professional communication.

To summarise, the statement in the manuscript is based on the findings of a meta-analysis. For such an important component of the manuscript, one would expect references to original research. However the papers that the meta-analysis looks to provide minimal experimental support for VPs being able to develop clinical reasoning. Please could the authors either provide references to these experiments or make their statement "Overall, VPs are a suitable method for students to train their clinical reasoning abilities" far more speculative.

I hope these comments will be useful.

Author Response

Dear reviewer,

We would like to thank you again for your valuable feedback and the effort you put into the review of our manuscript.

Based on your comment, we replaced the reference by Cook et al. with more suitable and more recent studies showing indicators that VPs are suitable for learning clinical reasoning. In addition, we rephrased the sentence as recommended.

Many thanks and best regards.